# Emerging Direct Targeting β-Catenin Agents

**DOI:** 10.3390/molecules27227735

**Published:** 2022-11-10

**Authors:** Marianna Nalli, Domiziana Masci, Andrea Urbani, Giuseppe La Regina, Romano Silvestri

**Affiliations:** 1Laboratory Affiliated with the Institute Pasteur Italy—Cenci Bolognetti Foundation, Department of Drug Chemistry and Technologies, Sapienza University of Rome, Piazzale Aldo Moro 5, 00185 Rome, Italy; 2Department of Basic Biotechnological Sciences, Intensivological and Perioperative Clinics, Catholic University of the Sacred Heart, Largo Francesco Vito 1, 00168 Rome, Italy

**Keywords:** β-catenin, inhibitor, small molecule, anticancer agent

## Abstract

Aberrant accumulation of β-catenin in the cell nucleus as a result of deregulation of the Wnt/β-catenin pathway is found in various types of cancer. Direct β-catenin targeting agents are being researched despite obstacles; however, specific β-catenin drugs for clinical treatments have not been approved so far. We focused on direct β-catenin targeting of potential therapeutic value as anticancer agents. This review provides recent advances on small molecule β-catenin agents. Structure-activity relationships and biological activities of reported inhibitors are discussed. This work provides useful knowledge in the discovery of β-catenin agents.

## 1. Introduction

β-Catenin is a key multiplayer protein of the Wingless/integrase-1 (Wnt)/β-catenin signaling pathway [1,2]. The Wnt-signaling pathway plays key roles in regulating cell fate, proliferation, tissue homeostasis and maintenance and embryogenesis [3,4,5]. Without a Wnt ligand, β-catenin is phosphorylated in the cytosol by the destruction complex. After phosphorylation, β-catenin dissociates from the complex and undergoes ubiquitination by β-transducing repeats-containing proteins (β-TrCP) and then degradation by proteasome. Alternatively, β-catenin undergoes phosphorylation by the destruction complex and then is ubiquitinated by the β-TrCP, the result of which is a minimal amount of β-catenin remaining in the cells [6] (Figure 1, left panel) Upon binding with a Wnt ligand, the interaction with Frizzled (Fzd) receptors and low-density lipoprotein receptor-related proteins 5 and 6 (Lrp5/6) triggers downstream events involving the recruitment of Dishevelled 1 (DVL-1) and Axin, following which β-catenin dephosphorylates and translocates into the nucleus. Accumulation of β-catenin in the nucleus prompts the expression of T-cell factor (Tcf) and the lymphoid enhancer-binding factor (LEF) transcriptional factors. It recruits transcriptional coactivators, such as CREB-binding protein (CBP)/p300 [7]; B-cell lymphoma 9 (BCL9) [8,9] and its paralogue BCL9-like (BCL9L) [10]; and Pygopus (Pygo 1 or Pygo 2) [11], and induces epigenetic modifications [12] (Figure 1, right panel). β-catenin may also bind to E-cadherin in the cytoplasm to recruit actin filaments [13]. The Wnt/β-catenin pathway is found hyperactivated in cancer. The accumulation of β-catenin in the cell nucleus triggers mutations, driving the transcription of oncogenes, such as Jun, Axin-2, c-Myc and *Cyclin-D1*, resulting in the initiation and progression of various types of carcinoma, including colon cancer, hepatocellular carcinoma, pancreatic cancer, lung cancer and ovarian cancer [14,15].

Various inhibitors of the Wnt/β-catenin pathway have been reported to date. These compounds target the Wnt/β-catenin pathway through different mechanisms: Wnt ligand, its receptor, β-catenin subcellular localization or the β-catenin transcriptional complex. The therapeutic potential of targeting the Wnt/β-catenin pathway has been reported in recent excellent reviews [16,17]. Specific drugs against this signaling pathway for clinical treatments have not been approved so far. In this work, we review recent progress in the development of direct β-catenin inhibitors, with particular emphasis on small molecules. Development, structure–activity relationship (SAR) and biological activity are reported.

## 2. Inhibitors the Wnt/β-Catenin Pathway

### 2.1. FH535

A series of sulfonamides were discovered by the high-throughput screening (HTS) of a library of 11,600 low-molecular-weight compounds by Handeli and Simon at the Cancer Research Center, Seattle, Washington [18]. For the high-throughput screen (HTS), the authors used a cell-based β-catenin/Tcf-responsive reporter containing either an optimized or mutated Tcf-binding element that was stably transfected into HepG2 cells expressing high levels of nuclear β-catenin. Compound FH535 (**1**), characterized by the presence of a nitro group at position 4 of the B ring, showed the strongest β-catenin/Tcf inhibition. Due to the structural similarity (carboxamide in place of the sulfonamide group) to the peroxisome proliferator-activated receptor isotype γ (PPARγ), antagonists GW9662 (**2**) and T0070907, compound **1** was tested for the ability to inhibit the PPARδ and PPARγ. In fact, β-catenin may improve PPARγ activity, and PPARγ regulates the β-catenin/Tcf interaction [19,20], indicating a potential mechanism of cross-talk between the Wnt and the PPAR-signaling pathways. Compound **1** was selectively toxic to some colon, lung and liver carcinomas expressing high or active Wnt/β-catenin pathways with LC_50_ values in the low micromolar range, but not to cells in which the Wnt/β-catenin signaling was not active. As an inhibitor of the HCC15 lung adenocarcinoma cells, **1** showed the strongest potency with LC_50_ of 3.5 μmol/L. The SW48 and HCT116 colon carcinoma cells were resistant to **1** at LC_50_ > 30 μmol/L. Compound **1** inhibited the Wnt/β-catenin pathway and antagonized both PPARγ and PPARδ. In contrast, compound **2** was specifically antagonistic to PPARγ and was unable to antagonize PPARδ and to inhibit the Wnt/β-catenin pathway. As an explanation, **2** requires Cys285 to covalently bind PPARγ [21], whereas the antagonistic activity of **1** does not depend on this cysteine residue. In fact, the activity of **1** was dependent on the presence of PPAR agonists inside the cell that can antagonize its activity. In this study, compound **1** arose as a useful tool to analyze the interactions between the Wnt/β-catenin and the PPAR signaling pathways.

### 2.2. Analogs of FH535

Kril and co-workers conducted structure-activity relationship (SAR) studies of **1** by replacing the 2,5-dichlorophenyl group with dihalogen patterns [22]. Replacing the 2,5-dichlorophenylsulfonyl substituent in **1** with a 2,6-difluorosulfonyl or 2,6-dichlorosulfonyl pattern, Kril generally obtained derivatives more active than **1** in the β-catenin/Tcf-dependent assay (luciferase-based TOPFlash assay) as well as on [^3^H]-thymidine incorporation assay. Compounds **3** and **4**, but also the non-halogenated derivative, **5,** demonstrated activity two-fold superior to compound **1**; other SAR modifications did not provide significant improvement over **1**. The authors hypothesized the involvement of a variety of targets and observed partial agreement between the [^3^H]-thymidine incorporation and the TOPFlash assays: for example, compound **3** was superior to **1** in the [^3^H]-thymidine incorporation assay but showed weak activity in the TOPFlash assay; on the contrary, methyl benzoate **6** was weakly active in the [^3^H]-thymidine incorporation assay but was comparable to **1** in the TOPFlash assay.

### 2.3. MSAB

High-throughput screening was conducted by Hwang’s group to identify inhibitors capable of modulating β-catenin-mediated Wnt signaling in HCT116 cancer cells. The authors conducted the HTS utilizing HCT116 cells with a mutant allele of β-catenin lacking Ser45, the phosphorylation target of CK1α, thus leading to high levels of active b-catenin. These studies led them to identify MSAB (**7**) as an inhibitor of the Wnt/β-catenin signaling pathway [23]. Compound **7** inhibited Tcf luciferase reporter activity in HCT116 cells with luciferase-based reporter containing Tcf binding sites (TOP-Luc), while it showed little effect on FOP-Luc. Compound **7** decreased the cell viability of Wnt-dependent HCT116, HT115 and H23 cells and inhibited tumor growth in a xenograft mice model after 2 weeks’ treatment at 10–20 mg/kg. In these studies, **7** proved to bind to β-catenin in a concentration-dependent manner, promoted its degradation in a proteasome-dependent manner, and specifically downregulated Wnt/β-catenin target genes in DLD-1, SW480 and LS174T cancer cell lines. It was suggested as a potential therapeutic agent for the treatment of tumors dependant on the Wnt/β-catenin signaling pathway.

### 2.4. Analogs of MSAB

Nalli and co-workers [24] found inconsistency in the SARs reported by Kril [22] and Hwang [23]. In particular, Nalli’s group observed that according to Kril [22], the dihalogenation pattern at the phenyl A was responsible for the effective inhibition of the Wnt/β-catenin signaling pathway, while according to Hwang [23], the presence of chlorine/fluorine atom(s) at position 4 of this ring weakened the inhibition of the Wnt/β-catenin signaling pathway. Moreover, substituents at position 4′ of the B ring of **7** were not explored. In a previous study, compound **8** was reported as a new β-catenin antagonist that was combined with NHERF1 PDZ1 domain inhibitors to enhance the CRC cell apoptotic response [25]. To further explore the SARs of this class of Wnt/β-catenin inhibitors, the group synthesized a small test set of analogs of **8** by changing the R_1_ substituents at the phenyl A and the position of the R_2_ ester functional groups at the phenyl B. Compound **9** was found to inhibit the effect on Wnt reporter with an IC_50_ value of 7.0 μM. It significantly reduced the c-Myc levels and inhibited HCT116 colon cancer cell growth with IC_50_ of 20.2 μM. Compound **9** did not violate the Lipinski and Veber rules and showed predicted Caco-2 and MDCK cell permeability (Papp > 500 nm s^−1^). Activation of c-Myc oncoprotein is one of the leading events of tumorigenesis and a high level of c-Myc correlates with the tumor’s survival. Direct or indirect suppression of c-Myc may slow down or reverse the cancer growth rate [26,27]. Since c-Myc is an undruggable target, this work demonstrated the ability of Wnt/β-catenin pathway inhibitors to indirectly reduce c-Myc levels and consequently, cancer cell proliferation. Compound **9** was shown to have potential for the development of new anticancer therapies (Figure 2).

## 3. Inhibitors of β-Catenin /Tcf Interactions

### 3.1. LF3

Fang, et al. [28], performed AlphaScreen-based HTS of a library of 16,000 synthetic compounds from the central open access technology platform of ChemBioNet [29] using a Glutathione S-transferase (GST)-tagged [30] armadillo repeat domain of human β-catenin and a His-tagged *N*-terminal region of human Tcf4. The authors discovered five compounds, LF1 to LF5, that inhibited by 50% the β-catenin /Tcf4 interactions at concentrations <10 μM in both AlphaScreen and ELISA assays. In these assays, compound LF3 (**10**) showed IC_50_ of 1.65 and 1.82 μM, respectively, and was predicted with good plasma membrane permeability by Lipinski’s rule [31]. SARs studies showed that in the sulfonamide group (replacement with methyl-, fluorine-, or nitro-eliminated inhibitory activity), the benzene tail and the length of the alkyl linker were essential for the activity. The distance between the benzene ring and the core structure was also crucial; only analog **11**, bearing a benzyl group, exhibited inhibitory activity comparable to **10**. Compound **10** inhibited Wnt signaling by targeting the transcription complex and was shown to interfere with both β-catenin/Tcf4 and β-catenin/LEF1 in dose-dependent manners in protein extracts from HCT116 cells. Compound **10** was examined for disruption of the interaction between β-catenin and E-cadherin. In fact, common armadillo repeats [32,33] drive the interaction of β-catenin with Tcf4. Metastasis can arise from loss of function of β-catenin in adherens junctions [34]. The presence of E-cadherin and β-catenin colocalized at cell–cell adhesion sites was observed by immunofluorescence staining of MDCK cell. The amount of E-cadherin bound to β-catenin remained unchanged upon incubation of 3.3 to 60 μM **10**, with β-catenin immunoprecipitated from protein extracts of HCT116 cells. Compound **10** reduced cell motility, cell-cycle progression and the overexpression of many Wnt target genes (Bmp4, Axin2, survivin, Bambi and c-Myc). Compound **10** blocked the self-renewal capacity of cancer stem cells (CSCs), a significant result since high Wnt/β-catenin activity characterizes colon CSCs and other CSCs [35]. Compound **10** reduced tumor growth and induced differentiation of colon CSCs in a mouse xenograft model. The authors hypothesized that **10,** through the negatively charged sulfonamide group, forms binding interaction with Asp16 of TCF4, and thus to the Lys435 of the positively charged pocket of b-catenin. The hydrophobic cleft of the b-catenin formed by residues Cys466 and Pro463 might host the hydrophobic tail of **10**.

### 3.2. iCRT3

The Gonsalves research group set an assay for HTS to identify small molecule candidates that would induce phenotype in the absence of dAxin as a consequence of the loss of β-catenin or Drosophila Tcf (dTcf) [36]. The authors screened 14,977 small molecules from the collection of ICCB, Harvard Medical School, Boston, for their effect on modulation of dAxin-dsRNA-induced dTF12 reporter activity/CRT in Drosophila Cl8 cells. In the first screening, the authors identified 34 molecules (called iCRT, inhibitors of β-catenin responsive transcription) falling into three families: oxazoles, thiazoles, and thiazolidinediones. Among them, nine oxazoles showed significant inhibition on the activity of the dTF12-luciferase reporter gene. iCRT derivatives **12**, **13** and **14** weakened the interaction of β-catenin treated with inhibitor to the *N*-terminal domain of Tcf4 and significantly inhibited β-cat-Tcf interactions. Computational studies based on the β-cat crystal structure highlighted key contacts of β-catenin: Lys312 and Lys345 with Glu16 on Tcf; Lys435 with Asp16 on Tcf; and Phe253 and Phe293 with Leu48 on Tcf [37,38]. Mutations in Lys312 and Lys435 weakened the transcription activation of the luciferase reporter TOPFlash by β-catenin [39]. The oxazole derivative iCRT **12** was significantly superior to the previously reported inhibitor Calphostin C, a PKC inhibitor that also antagonizes the Tcf/β-catenin complex with a similar inhibitory mechanism. Compounds **12**–**14** showed potent inhibition in the luciferase-based Wnt reporter assay, disrupted the interaction of β-catenin with Tcf4 without affecting its interaction with E-cadherin and α-catenin, inhibited the transcription of the downstream target genes of β-catenin, such as WISP-1, Axin-2, CycD1, and c-Myc and a variety of Wnt-responsive phenotypes. iCRTs **12**–**14** had an efficacy comparable with fluorouracil (5-FU) in human CRC. Computational studies revealed that **12** and **13** form highly specific electrostatic contacts with Lys435 and Arg469 of TCF and β-catenin. The carboxamide NH of compound **12** forms a hydrogen bond with Arg469 on β-catenin, a residue that is critical for stabilizing β-cat-TCF4 interactions. The carboxylic group of **13** mimics interactions mediated by the side chain of Asp16 on TCF4 with Lys435 on β-catenin [39].

### 3.3. Oxadiazolo-Pyrazino-Indole

Catrow’s group discovered six natural products of β-catenin/Tcf inhibitors though HTS of 52,000 compounds [40,41]. Optimization studies led them to identify the acyl hydrazone ZINC02092166 (**15**) [42], a new small-molecule inhibitor for the β-catenin/Tcf protein–protein interaction (PPI), using AlphaScreen and fluorescence polarization (FP) assays [43]. The AlphaScreen assay used *C*-terminally biotinylated human Tcf4 (residues 7–51) and *N*-terminally His_6_-tagged human β-catenin (residues 138–686) fragments. In the FP assay, the authors used fluorescein-labeled human Tcf4 at the *C*-terminal (residues 7–51) to form a complex with human β-catenin (residues 138–686), emitting polarized light. Compound **15** inhibited the TOPFlash luciferase activity in pcDNA3.1-β-catenin-transfected HEK293 and SW480 cells with the IC_50_ values of 0.86 and 0.71 μM, respectively, and the growth of SW480, HCT116 and HT29 cells with low micromolar IC_50_ values. However, **15** showed off-target effects at high concentrations due to the presence of both the acyl hydrazone moiety, a PAINS substructure [44], and the oxadiazolopyrazine ring. The hydrazide group of **15** was converted to an amide **16**. In the AlphaScreen and FP assays, **16** was similar to that of **15**, and V511, I569 and R469 were indicated as key residues by site-directed mutagenesis studies. Further optimization resulted in compound **17** yielding IC_50_ of 26 μM in the TOPFlash and FOPFlash luciferase reporter assay using pcDNA3.1-β-catenin-transfected HEK293 cells. Compound **17** blocked the transactivation of canonical Wnt signaling, suppressed the expression of Wnt-specific target genes and inhibited the growth of SW480 and HCY116 cancer cells with IC_50_ of 2.0 and 31 μM, respectively. The cell-based inhibitory activities correlated with the biochemical assays. The Kd value of the β-catenin/Tcf PPI suggested a significant selectivity over the β-catenin/cadherin and β-catenin/APC PPIs. Site-directed mutagenesis and SAR studies defined the binding mode of these inhibitors with β-catenin. These results showed that targeting the Tcf4 G^13^ANDE^17^ binding site of β-catenin with small molecules can lead to selective inhibition of the β-catenin/Tcf PPIs [42].

### 3.4. PNU-74654

Trosset, et al., performed docking studies of a 17,700-compound collection of Pharmacia at Nerviano Medical Sciences, Italy, using the QXP program [45]. The authors identified the binding site for small-molecule inhibitors near the K435/R469 region of β-catenin as the most promising hot spot for docking, which mostly contributes to the Tcf4/ β-catenin interaction. Of the original collection, 42 compounds were selected through criteria including association energy, contact energy, intramolecular ligand strain, Van der Waals repulsion, and visual inspection [46]. Of the 42 selected compounds, 22 underwent biophysical studies to determine binding to β-catenin and competition with Tcf4. The WaterLOGSY NMR screening, a method that finds out the compound-macromolecule interactions [47,48], led Trosset to identify seven binders of β-catenin. In the isothermal calorimetry (ITC) experiments, two active compound mixtures contained three of the seven hits detected by the NMR screening effort and showed about a 10-fold reduction of Tcf4-binding affinity. From the NMR and ITC screen, PNU-74654 (**18**) was identified as a Tcf4 competitor. It showed K_D_ of 450 nM in direct-binding ITC experiments and specific inhibition for Tcf-4 transactivation in the cellular luciferase reporter system. Binding modes were proposed for **18** and two analogs identified through a similarity search. Analog 1, with decreased affinity, lacked the methyl group at position 2 of the furane, and analog 2, with a piperidine moiety in place of the distal phenyl group that was not a competitor with Tcf 4. The proposed binding mode of **18** to the Lys435/Lys469 hot spot of β-catenin highlighted two pockets, with Asp16 of Tcf4 and with the groove of β-helices 23 and 30 of the armadillo repeat. One of the pockets contains the methylfurane where the methyl group laid in a narrow cleft, whereas the other pocket contains the phenyl moiety [45].

### 3.5. UU-T01

Researchers at the University of Utah [49] identified three key hot spots on β-catenin for binding to Tcf (K435/K508 for D16/E17 of Tcf4; and K312/K345 for E24/E29 of Tcf4) through two alternative conformations and a hydrophobic pocket lined for F253, I256, F293, A295, I296, V44 and L48 of Tcf4 [38,50,51]. The authors conducted alanine scanning and surface plasmon resonance (SPR) experiments to quantify the contribution of each hot spot region and derive the key elements of β-catenin for binding to Tcf. A bioisosteric replacement approach was used to design carboxylic acid bioisosteres that better match the critical binding elements. The inhibitory activities of six compounds toward disrupting β-catenin/Tcf4 PPIs were determined by the FP assay [43]. The most potent inhibitor UU-T01 (**19**) exhibited a Ki of 3.14 μM as a disruptor of β-catenin/Tcf interactions. In the ITC assay, **19** showed Kd of 0.531 μM to wild-type β-catenin, and in the AlphaScreen assay, displayed a Ki of 7.60 μM. Both the FP and AlphaScreen assays indicated that **19** completely disrupted β-catenin/Tcf4 PPIs. The ITC indicated Lys435 of β-catenin is critical for binding to **18**. The Kd values, Arg469Val of 3.21 μM (a mutation does not affect the binding of β-catenin to Tcf4) and Lys508Ala of 1.59 μM, showed that these residues involved in the binding of 18 to β-catenin (Figure 3).

### 3.6. UU-T02/03

Alanine scanning and biochemical assays led Huang and co-workers to identify a selective binding site that can differentiate β-catenin/Tcf, β-catenin/cadherin and β-catenin/APC interactions [52]. Comparing the crystal structures of complexes of β-catenin with Xenopus Tcf, human Tcf4, mouse Lef1 E-cadherin and APC, the authors observed that β-catenin uses the same armadillo repeats to bind Tcf (human Tcf4 residues 7–51), cadherin (human E-cadherin residues 819–873) and APC (residues 1477–1519 of human APC 20-amino acid repeat 3, APC-R3). The binding modes of Tcf, cadherin and APC in this region were reported to be identical and the binding modes to β-catenin to be mutually exclusive [33,37]. However, the authors observed subtle but very important differences of the binding modes in the crystal structures of the sequences flanking Tcf4 D16, E-cadherin D830 and APC-R3 D1486, and in particular, the binding site between armadillo repeats 9 and 10 of β-catenin. Tcf4 G^13^ANDE^17^, owning four well-defined pockets, binds to this site. New β-catenin/Tcf inhibitors were designed based on a peptidomimetic strategy; the G^13^ANDE^17^ sequence of human Tcf4 was used as the starting template. In site-directed mutagenesis studies and the FP assay, UU-T02 (**20**) exhibited a Ki value of 1.32 ± 0.56 μM against the wild type β-catenin and completely disrupted the β-catenin/Tcf PPIs. The carboxylic acid **20** exhibited dual selectivity for β-catenin/Tcf over β-catenin/cadherin and β-catenin/APC interactions. The corresponding ethyl ester UU-T03 (**21**) easily crossed the cell membrane and inhibited β-catenin/Tcf PPIs in a dose-dependent manner but had no effect on β-catenin/cadherin and β-catenin/APC PPIs. In the Wnt-responsive luciferase reporter assay, **21** yielded IC_50_ of 28.7 and 37.6 μM in SW480 and Wnt-activated HEK293 cells, and in the MTs assay on the growth of SW480, HCT116 and HEK293 colorectal cancer cells, it yielded IC_50_ values of 10.8, 11.0 and 84.0 μM. Site-directed mutagenesis studies to evaluate the binding mode of **20** with β-catenin highlighted the importance of the hydrophobicity of pocket B of the Tcf4 G^13^ANDE^17^ binding site for inhibition. The guanidino group of Arg474 was found to be more important than that of Arg515 for binding with **20**. The deletion of two gaunidino groups of Arg474 and Arg515 remarkably reduced the β-catenin inhibitory activity of **20**.

### 3.7. HI-B1

Shin’s group designed benzimidazole HI-B1 (**22**) [53] as a cyclic derivative of resveratrol based on the previously reported effect of resveratrol against the Wnt/β-catenin pathway [54]. In DLD-1 and Caco-2 cell lines, compound **22** inhibited β-catenin/Tcf4 luciferase activity and decreased mRNA expression of Cyclin D1 and Axin2, while it attenuated H-ras/β-catenin-induced foci formation. The direct binding of **22** with β-catenin disrupted the interaction between β-catenin and Tcf4 in vitro. Compound **22** decreased the growth of a patient-derived xenograft (PDX) colon cancer with a high expression level of β-catenin. The key role of the nitrogen atom at position 3 of the benzimidazole ring for the interaction of **22** with β-catenin was predicted by molecular modeling studies and confirmed in the DLD-1 cell proliferation and β-catenin/Tcf4 luciferase assays. The nitrogen at position 3 of the benzimidazole forms a hydrogen bond with β-catenin and its replacement with a carbon atom decreased the activity of **22** against DLD1 colon cancer cell growth.

### 3.8. PFK Compounds

Lepourcelet screened approximately 7000 purified natural compounds from proprietary and public collections to identify compounds that disrupt f/β-catenin complexes [40]. The authors developed a binding assay for HTS. Purified β-catenin (amino acids 134–668) was incubated sequentially with a Tcf4 fragment (residues 8–54) fused to GST and an anti-GST antibody. The authors identified eight compounds that showed dose-dependent inhibition of the Tcf4/β-catenin interaction with IC_50_ < 10 μM: PKF115-584, CGP049090 and PKF222-815 were isolated from fungal organisms; PKF118-744 (**23**), PKF118-310 (**24**) and ZTM000990 (**25**) originated in Actinomycete strains; NPDDG39.024 was isolated from the marine sponge Fascaplysinopsis reticulata; and NPDDG1.024 was derived from NCI collections. In the ELISA, compounds **15**–**17** inhibited the Tcf4/β-catenin association by 50% with IC_50_ values of 2,4, 0.8 and 0.64 mM, respectively. The selected compounds showed antagonism to the β-catenin-dependent activities, including the expression of c-Myc or Cyclin D1, cell proliferation, and duplication of the Xenopus embryonic dorsal axis. The authors report that molecular mechanisms by which the isolated compounds interfere with the targeted interaction are unclear. In multiple myeloma (MM), PKF115-584 (**26**) blocked expression of Wnt target genes and induced cytotoxicity in both patient MM cells and MM cell lines without a significant effect in normal plasma cells. In xenograft models of human MM, **18** inhibited tumor growth and prolonged survival [55].

### 3.9. Henryin

Henryin (**27**) is an ent-kaurane diterpenoid isolated from Isodon rubescens var. lushanensis, a plant used in folk medicine. Li reported that **27** selectively inhibited the growth of human CRC cells with a GI_50_ value in the nanomolar range. The mechanism of action of **27** was to antagonize the Wnt-signaling pathway. Compound **27** reduced the expression of Cyclin D1 and c-Myc, and induced G1/S phase arrest in HCT116 cells by impairing the association of β-catenin/Tcf4 transcriptional complex, likely through directly blocking the binding of β-catenin to Tcf4 [56]. Compound **27** did not affect the stabilization of β-catenin or its nuclear translocation. It strongly inhibited the binding of TCF4 to β-catenin in a dose-dependent manner in SW480 cells and in the in vitro binding assays with purified recombinant β-catenin and TCF4 proteins.

### 3.10. BC21

Tian’s group performed virtual screening studies [57] using the molecular modeling software Autodock [58] of a library of 1990 small molecules from the NCI open database. The structure of the receptor was extracted from the β-catenin/Tcf4 complex crystal structure and the N-terminal moiety was truncated to facilitate the preprocessing of the receptor. Three key “hot spots” (sites A, B, and C) on the β-catenin for the binding of Tcf4 were identified based on the literature report [59]. The authors investigated 100 compounds using a cell-based TOP/FOP assay for testing the β-catenin/Tcf4 signaling in HCT116 cells that were transfected with a luciferase reporter gene (luciferase). As a result, BC21 (**28**) was identified as a dose-dependent inhibitor of the β-catenin/Tcf4-driven luciferase activity in both primary HCT116 and pcDNA3.1-β-catenin-transfected HEK293 cell lines. Compound **28** decreased the HCT116 cell viability, at 5 μM, it inhibited > 80% colony forming activities and was not cytotoxic on normal HEK293 and HUVEC cells. Site A was sufficiently wide to adopt a small molecule that could disrupt β-catenin/Tcfs interactions. Lys435, Arg469, Lys508, Arg515 and Glu571 were key polar residues for binding of β-catenin with Tcf4. Important interactions on site A were Asp16 of Tcf4 with Lys435 and Asn430; Glu17 of Tcf4 with Lys508; and Ile19 and Phe21 of Tcf4 side chain with Arg386 aliphatic portion of β-catenin. Compound **28** was close to the Ile19 and Phe21 residues of Tcf4, and formed hydrophobic interactions with Pro463 ring of the β-catenin hydrophobic cleft. The other half of **28** laid close Asp16, also interfering with the β-catenin/Tcf4 interaction (Figure 4).

### 3.11. aStAx-35

StAx peptides were investigated for the ability to compete with TCF4 for binding of β-catenin by means of an immobilized GST-tagged CBD of TCF4(1-52). Stapled peptides aStAx-35 (m.w. 788.8) (**29**) and aStAx-35R (m.w. 798,1) (**30**) inhibited competitively the binding of β-catenin to GST-TCF4(1-52) in the presence of a N-terminal acetylation. The crystal structure of N-terminally acetylated 29 in complex with residues 134–665 of β-catenin was resolved at 3.0 Å resolution. The β-catenin/**29** stapled peptide crystal structure was deposited under PDB 4DJS. The binding site on β-catenin of **29** was revealed using the unliganded β-catenin structure PDB 1QZ7. The Trp residues at positions 468 and 481 increased the binding affinity for β-catenin engaging recognition pockets of the surface of β-catenin. Trp481 formed interactions with Trp338 of β-catenin and contributed to a H-bond with the carboxyl oxygen of Asp299 in β-catenin. Compound **30** proved to bind at the same Axin site but did not increase steady-state levels of β-catenin. Both **29** and **30** suppressed luciferase activity; the proliferation of DLD1 and SW480 cells was blocked by **30**. The panel of canonical Wnt-driven genes was strongly suppressed, indicating that **30** antagonizes the nuclear form of β-catenin and hampers transcriptional coactivation for TCF proteins [60].

## 4. Inhibitors of β-Catenin to BCL9

### 4.1. Carnosic Acid

De La Roche and co-workers identified some natural compounds, including the carnosic acid (**31**) from rosemary, that inhibited, in a dose-dependent manner, the binding of β-catenin to BCL9 in vitro, and β-catenin-dependent transcription in CRC cells [61]. Biophysical analysis highlighted the labile α-helix (H1) at the amino terminus of the β-catenin Armadillo repeat domain bordering the BCL9-binding site, as a crucial element for the activity of **31**. The mechanism of action of **31** was to promote selectively the proteasomal degradation of unphosphorylated β-catenin in an H1-dependent manner. To identify inhibitors of this interaction, the authors developed an in vitro assay that monitors the binding of His-HD2 (BCL9 homology domain 2) to glutathione S-transferase (GST)-ARD (immobilized on glutathione-coated microplates), using a colorimetric assay to quantify bound His-HD2 after addition of compounds. NMR saturation transfer difference (STD) spectroscopy was used to investigate the interaction of **31** with its target domain using the minimal HD2-binding domain within its N-terminus (four repeats: R4) [62]. Evidence highlighted that the effects of **31** on R4 could explain its in vivo effects on β-catenin. The authors proposed that the metastable H1 also predisposed β-catenin to low-grade aggregation in vivo, and that this is exacerbated by **31**, which could earmark β-catenin for proteasomal degradation. Saturation transfer difference (STD) NMR spectroscopy assays were performed to indentify the binding domain of **31** with purified R4 or BCL9 homology domain 2 (HD2). R4 tested positive in this ligand-observed binding assay whereas HD2 was negative. Titration with varying concentrations of R4 allowed researchers to estimate a Kd in the 5–20 μM range. The STD assays unequivocally identify R4 as the molecular target of **31**.

### 4.2. Bispyrrodinylium-Carboxamide

Hoggard and co-workers designed and synthesized novel small-molecule inhibitors that selectively disrupted β-catenin/BCL9 over β-catenin/cadherin PPIs [63]. The binding mode of new inhibitors was characterized by site-directed mutagenesis and SAR studies. The new inhibitors suppressed the transactivation of canonical Wnt signaling, downregulated the expression of Wnt target genes, and inhibited the growth of Wnt/β-catenin-dependent cancer cells. The authors considered two hot regions at the β-catenin/BCL9 interface: in hot region one, residues D162, E163 and D164 of human β-catenin interact with residues H358 and R359 of human BCL9; in hot region two, residues L366, I369, and L373 of BCL9 interact with a hydrophobic pocket bordering with residues L159, V167, L160, A171 and M174. Compound **32** was provided with two positively-charged pyrrolidino groups and a substituted phenyl ring were introduced to improve the selectivity between α-helix-mediated PPIs. AlphaScreen and ITC studies consistently showed that **32** bound wild-type β-catenin but not BCL9. Compound **32** inhibited the β-catenin/BCL9 PPI, but not β-catenin/E-cadherin PPI, in colorectal SW480, HCT116, HT29 CRC cells and triple-negative breast cancer cells MDA-MB-231 and MDA-MB-436, all overexpressing the Wnt signaling with IC_50_ values in the low micromolar range. In the AutoDock predicted the binding conformation within β-catenin (PDB 2GL7); main contacts were the H-bonds of the protonated pyrrolidino groups with D145 and E155, the interactions of the mono and difluoro substituted phenyl rings with L178 and L159, respectively, and the contact of the oxygen atom linked to one pyrrolidino group with L156.

### 4.3. hsBCL9CT-24

Feng’s group designed peptides based SAH-BCL9B, a previously reported peptide of the BCL9 homology domain 2 (BCL9-HD2) with promising inhibition of the Wnt pathway [64]. To improve activity, the authors focused on stapled peptides and conducted lead optimization studies using full-length BCL9-HD2. Among the peptides synthesized at AnaSpec, hsBCL9CT-24 (**33**) showed the most potent in vitro activity and exhibited stronger binding affinity to β-catenin compared to BCL9-HD2A. In the homogeneous time resolved fluorescence (HTRF) binding assay, hsBCL9CT-24 yielded a Kd value of 4.21 nM, compared to Kd value of 192.3 nM of SAH-BCL9B. In a modified AlphaScreen that allows greater surface area of β-catenin/BCL9 binding [65,66] to determine the potency of inhibitors in disrupting the β-catenin/BCL9 interaction, hsBCL9CT-24 showed a low Kd value of 4.73 nM. hsBCL9CT peptides were found to be highly permeable and readily taken up by treated cells. hsBCL9CT-24 was superior to SAH-BCL9B as inhibitor of β-catenin transcriptional activity in HCT116 and Colo320DM (dependent on β-catenin and BCL9) cell lines. hsBCL9CT-24 effectively reduced tumor growth in the Colo320DM xenograft and in PDX models of human CRC, showed a favorable pharmacological profile and minimal toxicity. Docking of **33** into the β-catenin hydrophobic pocket (PDB 3SL9) performed by GlideXP Maestro Schrodinger highlighted binding interactions with Ala187, Asp145, Leu178, Met174, Lys170 and Val 167 amino acids in the β-catenin hydrophobic pocket (Figure 5).

## 5. Conclusions

Great efforts are underway to discover direct inhibitors of β-catenin as anticancer agents. The direct inhibition of the oncogenic β-catenin represents an intriguing approach to ameliorate the efficacy and tolerability of the anticancer therapy, compared to upstream effectors of the Wnt/β-catenin signaling pathway. However, various difficulties hinder the progression to clinical studies of these agents, including the lack of co-crystal structure of small molecules in complex with β-catenin, the suboptimal metabolic and physiochemical profile of the known β-catenin inhibitors and the emergence of unwanted off-target effects. Continuous efforts to develop inhibitors of the Wnt/β-catenin signaling pathway highlight the potential of this strategy for the treatment of cancer. The drug discovery on novel β-catenin agents urgently needs information of binding sites, binding modes, and selectivity over other proteins. Beyond that, we should consider the complexity of the cancer development involving cross-talk between cellular signaling pathways, multiple factors and gene mutations. Advancement in structural information of β-catenin as well in knowledge about the Wnt pathway and the cross-talk mechanisms involved in normal and pathological conditions, we are confident, will pave the way for the development of effective β-catenin-targeting anticancer agents. Biological assays used to evaluate the β-catenin activity and activities of compounds **1**–**33** are summarized in Appendix A. The agents described in this review provide a valuable basis for the development of direct β-catenin inhibitors.

## Figures and Tables

**Figure 1 molecules-27-07735-f001:**
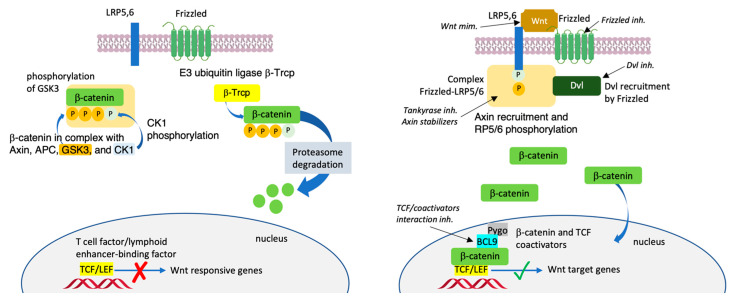
Sketch of the canonical Wnt/β-catenin signaling pathway without a Wnt ligand (left panel) and upon binding with a Wnt ligand (right panel). Molecules being evaluated in clinical studies: Wnt ligand mimetics, inhibitors of Frizzled, Dvl, Tankyrase, TCF/coactivators interactions or Axyn stabilizers.

**Figure 2 molecules-27-07735-f002:**
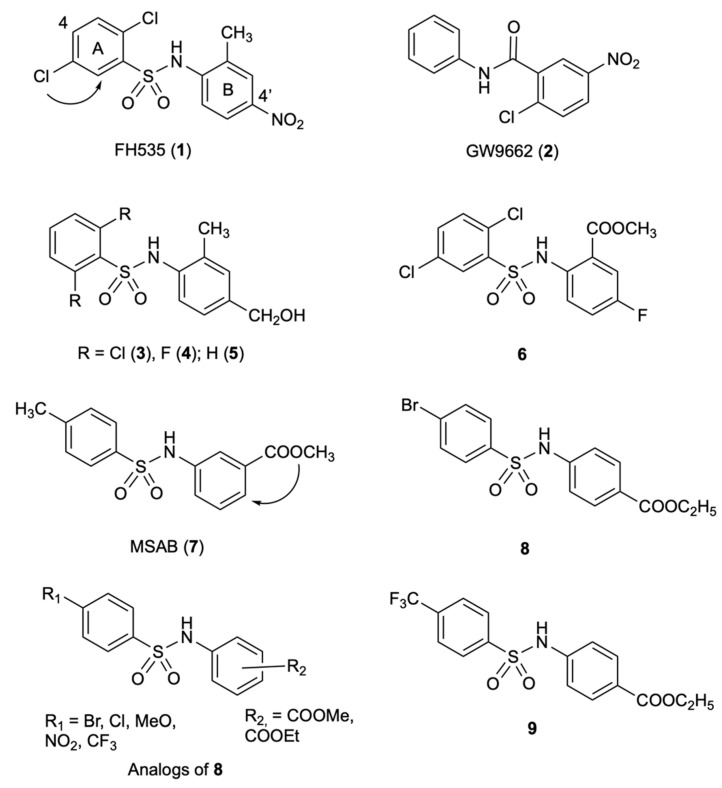
Chemical structures of sulfonamides **1**–**9** inhibitors of the Wnt/β-catenin pathway.

**Figure 3 molecules-27-07735-f003:**
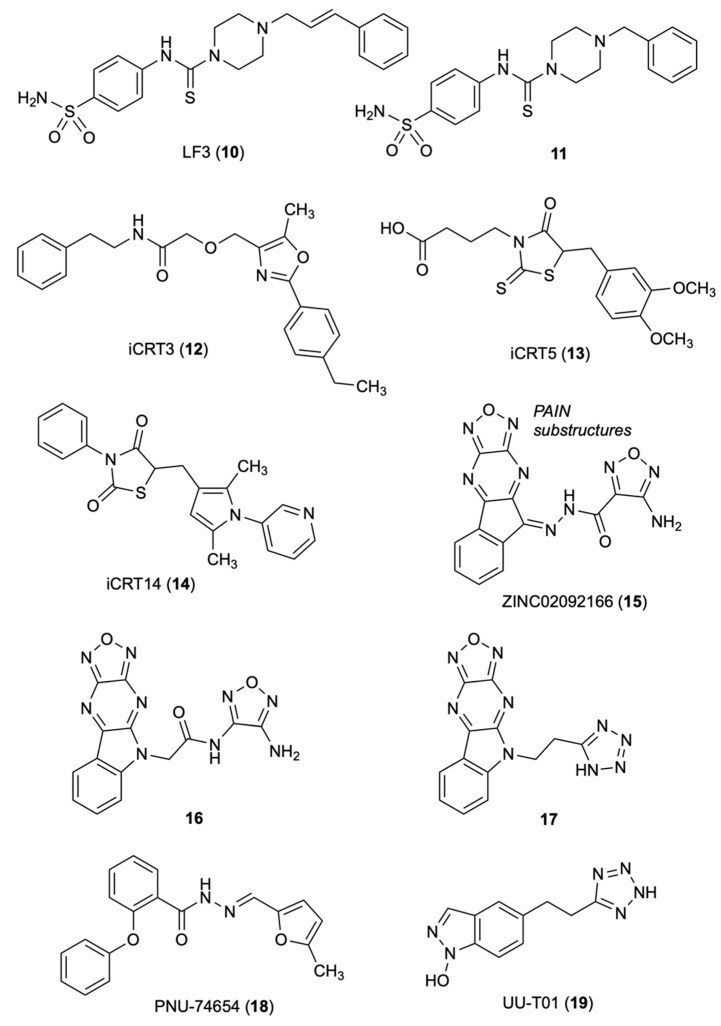
Chemical structures of compounds **10**–**19** inhibitors of β-catenin/Tcf interactions.

**Figure 4 molecules-27-07735-f004:**
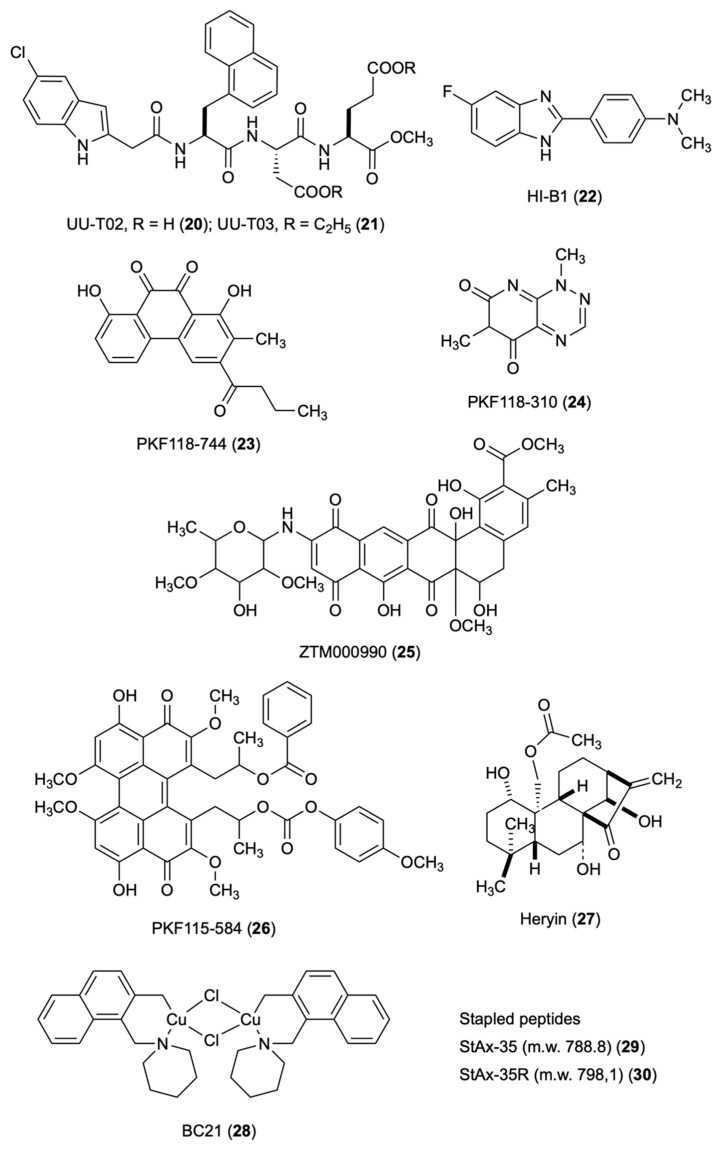
Chemical structures of compounds **20–30** inhibitors of β-catenin/Tcf interactions.

**Figure 5 molecules-27-07735-f005:**
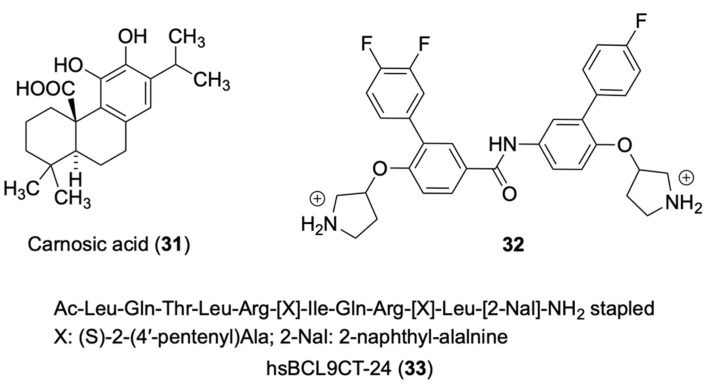
Chemical structures of compounds **31**–**33** inhibitors of β-catenin to BCL9.

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
