# Peer review of "Emerging Direct Targeting β-Catenin Agents"

_molecules, 2022, doi:10.3390/molecules27227735_

Round 1
Reviewer 1 Report
The manuscript is a complete review providing recent advances on small molecule β-catenin agents. In general, the review will be of interest to those working in cancer drug design and specifically to those interested in targeting β-catenin. However, there are some points to be considered.
1.- It would be desirable to include a figure showing the Wnt/ β -catenin pathway according to the information mentioned in the text.
2.- Figures are not cited in the text. Additionally, in figures 1 and 2, some parts of compounds structure are highlighted, but nothing is described about it in figure legends.
3.- In conclusion section, the point of view of the authors about the future of β-catenin agents design should be included, not only general aspects.
Author Response
We thank the Reviewer 1 for comments and suggestions which has enabled us to revise and improve the manuscript. In the attached file "Response to the Reviewer 1" we explain point by point how we have dealt to the comments and suggestions the Reviewer 1. We hope to receive a positive response for publication in the Molecules.

Reviewer 2 Report
This review summarized the direct β-catenin targeting agents as the emerging cancer therapeutic strategy. The development of inhibitors, biological activities and structure-activity relationships of the reported inhibitors are discussed, inspiring the further drug design and development of direct targeting β-catenin agents. The topic is very interesting and fits the scope of the journal. In general, the manuscript is well-organized, and the references can support the conclusions. However, some key issues are required to be addressed before its publication on Molecules.
1. The direct targeting β-catenin inhibitors are mainly such inhibitors blocking interaction of β-catenin with the other critical proteins. The inhibitors sections of this manuscript were suggested to be re-organized according to the inhibition of different protein-protein interactions, and then the structure features of these inhibitors (as the secondary classification criteria).
2. The characteristics (including binding pocket, if available) of the key protein-protein interactions involving β-catenin are suggested to be described, which is very important for the targeted drug discovery.
3. The developed/used biological assays in the evaluation of direct targeting inhibitors of β-catenin are suggested to be summarized in a separate section, to give readers more ideas about discovery and development of direct targeting β-catenin inhibitors.
4. The stapled peptide StAx-35 directly binds to β-catenin and inhibits the oncogenic Wnt signaling (Ref.: Proc Natl Acad Sci U S A. 2012;109(44):17942‐17947). As the only co-crystal structure of β-catenin in complex with its inhibitor (PDB: 4DJS) (even though it is a peptide), it is very important and is suggested to be described and discussed with a figure illustration in this manuscript.
5. Following the comment 4, the proposed binding pockets of the representative compounds (if available) are suggested to be included in this manuscript, and the possible reasons for the limited inhibitors and very limited crystallography study are suggested to be discussed.
6. The figure illustration of Wnt/β-catenin pathway with current therapeutic intervention is required in the introduction section to improve the readability of this manuscript.
7. The activity data are suggested to be listed along with the structures in the Figures, or summarized in a separate table, to improve the manuscript readability.
Author Response
We thank the Reviewer 2 for comments and suggestions which has enabled us to revise and improve the manuscript. In the attached "Response to the Reviewer 2" we explain point by point how we have dealt to the comments and suggestions the Reviewer 2. We hope to receive a positive response for publication in the Molecules.

Round 2
Reviewer 2 Report
After the authors’ revision according to my and others’ previous comments, the quality of this paper was significantly improved, and could reach the required quality standard for Molecules in my opinion. I suggest accepting it without further revision.